# Second-Generation Lignocellulosic Supportive Material Improves Atomic Ratios of C:O and H:O and Thermomechanical Behavior of Hybrid Non-Woody Pellets

**DOI:** 10.3390/molecules25184219

**Published:** 2020-09-15

**Authors:** Bruno Rafael de Almeida Moreira, Ronaldo da Silva Viana, Victor Hugo Cruz, Anderson Chagas Magalhães, Celso Tadao Miasaki, Paulo Alexandre Monteiro de Figueiredo, Lucas Aparecido Manzani Lisboa, Sérgio Bispo Ramos, Douglas Enrique Juárez Sánchez, Marcelo Carvalho Minhoto Teixeira Filho, André May

**Affiliations:** 1Department of Phytosanitary, Rural Engineering and Soils, School of Engineering, São Paulo State University (Unesp), Ilha Solteira, São Paulo 15385-000, Brazil; douglas.sanchez@valedoparana.com.br (D.E.J.S.); mcm.teixeira-filho@unesp.br (M.C.M.T.F.); 2Department of Plant Production, College of Agricultural and Technological Sciences, São Paulo State University (Unesp), Dracena, São Paulo 17900-000, Brazil; ronaldo.viana@unesp.br (R.d.S.V.); cruz.v.h@outlook.com (V.H.C.); ac.magalhaes@unesp.br (A.C.M.); celso.t.miasaki@unesp.br (C.T.M.); paulo.figueiredo@unesp.br (P.A.M.d.F.); lucas.lisboa@unesp.br (L.A.M.L.); sergio.bispo@unesp.br (S.B.R.); 3Brazilian Agricultural Research Corporation (Embrapa), Jaguariúna, São Paulo 13820-000, Brazil; andre.may@embrapa.br

**Keywords:** blending, cellulosic bioethanol, energy storage, fuel grade biosolids, natural binding agent, reinforcement, sustainable waste-to-energy technique

## Abstract

Pellets refer to solid biofuels for heating and power. The pellet’s integrity is of great relevant to ensure safe and effective transportation and storage, and comfort to stakeholders. Several materials that are supportive, whether organic and inorganic, to pellets exist. However, no work in the literature is linking making hybrid non-wood pellets with addition of residual biomass from distillation of cellulosic bioethanol, and this requires further investigations. Figuring out how effective this challenging agro-industrial residue could be for reinforcing non-wood pellets is accordingly the scientific point of this study focusing on management of waste and valorization of biomass. The pilot-scale manufacturing of hybrid pellets consisted of systematically pressing sugarcane bagasse with the lignocellulosic reinforcement at the mass ratios of 3:1, 1:1, and 1:3 on an automatic pelletizer machine at 200 MPa and 125 °C. Elemental contents of C and H, durability, and energy density all increased significantly from 50.05 to 53.50%, 5.95 to 7.80%, 95.90 to 99.55%, and 28.20 to 31.20 MJ kg^−1^, respectively, with blending the starting material with the reinforcement at 1:3. Preliminary evidence of residual biomass from distillation of second-generation bioethanol capable of highly improving molecular flammable/combustible properties, mechanical stability, and fuel power of composite non-wood pellets exist.

## 1. Introduction

The bioethanol can be from industrially processing sugary and starchy feedstock, such as the juice of sugarcane and corn’s grain [1]. However, if the bioethanol is from non-food feedstock, like lignocellulose, instead of from food-crops, it then falls rather in the strict category of second-generation instead of first-generation biofuels [2]. The production of second-generation bioethanol, or simply cellulosic ethanol, generally starts with the collection and transportation of lignocellulose to the plant. In the site, whether sugar-energy factory, biorefinery, or distillery, the feedstock streams up the major stages pre-treatment or preprocessing, enzymatic hydrolysis, fermentation, and separation/distillation [3]. The final stage of the process is to separate the ethanol from the broth through molecular sieves or hydrophilic membranes, then concentrate the product to the optimal fuel grade by distillation [4]. At downstream steps, burning down on residual biomass from separation/distillation can power upstream steps with heat and power. The re-use of this agro-industrial residue as an alternative biofuel technically is viable. However, it often is highly heterogenous in size and shape, lowly dense, highly hygroscopic, and poorly energetic. These disadvantages make in situ management, and transportation of residual biomass away from the station difficult and expensive. Development and implementation of cost-effective strategies to improve residual biomass to ensure cellulosic ethanol feasible to fabricate on an industrial scale and competitive with gasoline is, therefore, an urgent matter. The process of pelleting may be an option to enhance its handleability, transportability, and storability by reshaping it into strictly compact solid biofuels.

The pelleting is the technique of integrating principles of classical mechanics and environmental approaches to develop low-technical-quality raw material into fuel grade biosolids of great acceptance at the market [5]. Pellets can be from processing woody and non-woody biomass from agricultural, agro-industrial, foresty, and municipal segments [6]. Fuel pellets are replete with valuable benefits to rising world population. Economically, they sound as great storers and suppliers of sustainable and renewable energy to feed heat-and-power units. Complementarily, they are more effective and cheaper pathways to transport and store biomass in ships and trucks’ containers and indoor facilities, such as seasonal silos, warehouses, and rooms. Environmentally, firing on pellets, instead of firewood or coal and derivatives, can assist in both saving emission of greenhouse gases into the atmosphere and preventing woodlands from runaway deforestation. This is timely and of great importance to ensure cleaner production and safer ecosystems [7]. Socially, pellets can offer employment opportunities and extensively improve conditions of people living in rural zones, where access to national electricity grid is difficult [8].

The global production of wood pellets grew up substantially from 5 Mt in 2005 to about 30 Mt by 2017. The European Union (EU) is the world’s top producer and consumer of wood pellets. In the year of 2017, particularly, consumption of wood pellets in bags for heating in 500 kW boilers by EU’s residential customers from breached 9.5 Mt, at the price of €250 per metric ton. Meanwhile, South Korea and Japan account for the steadily rising Asian markets, thanks to governmental policies and programs, like the Feed-in Tariff and Renewable Portfolio Standard. Therefore, global production of wood pellets is likely to exceed 60 Mt by 2025 [9]. Wood pellets are, of course, the center of both production and consumption of this category of tiny solid biofuels worldwide. However, residues of softwood and hardwood, such as sawdust, chip, and bark, themselves would be not able to fulfill future demand for wood pellets [10]. Development of non-wood pellets is, therefore, challenging. Residual biomass from distillation of second-generation bioethanol may be an executable option. Household stoves, furnaces, boilers, and gasifiers can burn solid biofuels from agricultural and agro-industrial residues without any difficulty [11].

As far as we know, no work in the regular literature is linking the recycling of residual biomass from distillation of cellulosic ethanol with the production of pellets, whether pure and composite. In view of gaps and opportunities from the database, this scientific study purposes the systematically reshaping of this agro-industrial residue into an organic supportive material to develop high-performance hybrid non-wood pellets.

## 2. Results

### 2.1. Characterization of the Ingredients

#### 2.1.1. Physicothermal Properties

The ingredients were relatively similar in water content, but absolutely dissimilar in contents of volatile matter, fixed carbon, ash, C, H, and O (Table 1). The residual biomass was richer in fixed carbon, C, and H than sugarcane bagasse. Hence, its atomic ratios of C:O and H:O were higher. Both the materials ended up highly releasing ash from burning down.

#### 2.1.2. Distribution of Particles’ Size

The largest fraction of the mass of particles, irrespective of the material, was in the range of size of 0.425–0.5 mm (Figure 1). The residual biomass was slightly higher in the mass of finer particles of 0.15–0.25 mm. In contrast, particles for the sugarcane bagasse were visually coarser and generally more irregular in shape.

### 2.2. Effect of Supportive Material on the Quality of Hybrid Non-Wood Pellets

The blending significantly affected the pellets’ qualities (Table 2). The dataset was not sensitive to departures from normalcy referring to problems of machine, material, method, measurement, labor, and environment which might cause the process of pelleting to run out-of-control and, hence, drop the quality of the product. This supported the rigorous nature of the experiment and should ensure its adequacy and reproducibility to further investigations focusing on the subjects of optimization of converting residual biomass into an organic supportive material for making high-performance non-wood solid biofuels.

#### 2.2.1. Quality of Hybrid Non-Wood Pellets

Parametric estimations for the effect of addition on the diameter, durability, resistance to abrasion, HHV, and energy density; contents of water, fixed carbon, and ash; and contents of C and H were positive (Table 3). Values of *r*^2^ ranged as low as 0.645 for the sulfur content to upwards 0.995 for the ash content. Therefore, most of the first-power regression models accurately predicted the thermomechanical behavior of composite materials to go up proportionally with magnifying the systematic introduction of the residual biomass into the process of co-pelletization.

##### Physicomechanical

The use of the supportive material at the mass ratios of 1:3, 1:1, and 3:1, correspondent to 25, 50, and 75% total mixture, enabled the pellets to be 30.10, 27.80, and 2.570 mm in length, respectively (Figure 2). Pure pellets were longitudinally larger, 31.90 mm. Therefore, the reinforcement at the largest size of addition shortened the pellets by 19.45%, making them more geometrically compact and homogeneous.

Durability and resistance to abrasion in composite pellets were in the ranges of 97.10–99.95 and 90.70–92.55%, respectively, regardless of the size of addition. These variables of mechanical stability significantly were both lower in pellets purely consisting of sugarcane bagasse, 95.90 and 89.75%, respectively. Besides the improvement in geometry, the use of the agro-industrial residue as an alternative supportive material into the process of pelleting increased the durability and resistance to abrasion to upwards 4.05 and 1.10%, respectively. This enhancement assisted the pellets in more firmly supporting drops and shocks and, hence, reduced the loss of mass of dust and fines to the environment during handling, transportation and storage at non-reactive atmosphere.

##### Physicothermal

The residual biomass at 1:3, 1:1 and 3:1, persistently, enabled the composite pellets to be highly dense, 1405.25, 1440.25 and 1485.10 kg m^−3^, respectively (Figure 3). Pellets with no supportive material were lower in apparent density, 1375.15 kg m^−3^. Therefore, the potential solid additive at the largest size of blending significantly increased the pellet’s degree of compactness by 8.05%.

The addition of residual biomass at 1:3, 1:1, and 3:1 to the sugarcane bagasse, complementarily, caused the hybrid non-wood pellets to release larger amounts of thermal energy per mass and volume unit. The gross heat of combustion for these products was 20.70, 21.25, and 21.45 MJ kg^−1^, respectively. Pure pellets yielded approximately 20.50 MJ kg^−1^. Therefore, the reinforcement at the highest mass ratio of blending slightly increased the HHV by 4.65%. Composite pellets consisting almost entirely of supportive material consequently became the energetically densest, 31.80 GJ m^−3^, as they peaked in both the apparent density and HHV. These variables consistently were complementarily. Referential pellets were lower in energy density, 27.95 GJ m^−3^. Therefore, the reinforcement at 3:1 physically was enough to significantly increase this property by 13.75%.

##### Physicochemical


**1. Proximal**


Irrespective of the size of addition, composite pellets were 7.05–8.75% water, 70.05–76.40% volatile matter, 21.10–24.50% fixed carbon, and 2.65–5.45% ash (Figure 4). Pure pellets consisted of 6.70% water, 80.00% volatile matter, 19.10% fixed carbon, and 0.90% ash. Therefore, the residual biomass enabled the pellets to be relatively higher in contents of water, fixed carbon, and ash, but lower in volatile matter content. Practically, the use of the reinforcement significantly increased these proximal properties by up to 30.70, 25.95, and 505.55%, respectively, but decreased the volatile matter content by 4.70%. Blending impacted longer on the ash content.


**2. Elemental**


The composite pellets consisted of 52.60–53.55% carbon, 6.45–7.80% hydrogen, 38.85–40.45% oxygen, and 0.90–1.05% sulfur, regardless of the size of addition (Figure 5). Elemental composition of pellets purely consisting of sugarcane bagasse were 50.05% carbon, 5.35% hydrogen, 41.40% oxygen, and 2.45% sulfur. Therefore, the residual biomass enabled the pellets to higher contents of carbon and hydrogen and, of course, lower contents of oxygen and sulfur. The use of the reinforcement significantly increased the contents of hydrogen by up to 7.10 and 45.80%, but decreased the contents of oxygen and sulfur by up to 2.30 and 133.35%, respectively. Evidently, blending impacted longer on the sulfur content.

The highest contents of carbon and hydrogen and the lowest oxygen content enabled the pellets containing the supportive material at 3:1 as supplemental part of their composition the highest atomic ratios of C:O and H:O, 1.35 and 0.20 and, consequently, the lowest atomic C:H ratio, 7.50, respectively. Hence, the reinforcement assisted the composite pellets in concentrating rather carbon and hydrogen than oxygen. Expressive gains in carbon and hydrogen should improve flammability and combustibility of the fuel grade biofuel at the molecular scale.

### 2.3. Canonical Correlations between Characteristics of the Ingredients and Quality of Hybrid Non-Wood Pellets

The canonical correlation analysis described adequately the straight dependence of the pellet’s quality on the characteristics of the ingredients (graphical abstract). The canonical components, CI, CII, and CIII, collectively, tracked the most consistent thermomechanical performance from the hybrid materials almost entirely consisting of residual biomass. The explanation for the greatest physicochemical stability and fuel power of these products was the highest contents of water, fixed carbon, lignin, carbon, and hydrogen, as well as the largest availability of finer and more regular particles in the reinforcement at the largest size of addition. In contrast, the highest contents of extractive and oxygen, in combination with the highest mass of coarser particles were the most limiting properties of feedstock for both the physicomechanical stability and fuel power of pellets purely consisting of sugarcane bagasse. Practically, as long as the feedstock is richer in natural short-range and broad-range bonding mechanisms, as well as finer and more homogeneous in granulometry, the probability of optimization of both the pelleting throughout and development of dense, durable, and energetic non-wood pellets is large. Therefore, preliminary evidence of residual biomass from distillation of cellulosic ethanol capable of highly improving molecular flammable/combustible properties, physicomechanical stability, and fuel power of hybrid pellets exist.

## 3. Discussion

### 3.1. Characterization of the Ingredients

#### 3.1.1. Physicochemical Properties

To produce pellets sustainably, the feedstock must be low in volatile and high in fixed carbon, and its contents of water and ash must be no greater than 15% and 3%, respectively [12]. Otherwise, it can imply financial losses to the project and drop the pellet’s productivity, quality, and acceptance at the strictest potential markets. All the proximal properties of both the materials were in compliance with the critical ranges from the literature but ash. The very high ash content in both the ingredients may be resultant of mineral contamination due to dragging of sandy and soil during manipulation, transportation and storage in the sugar-energy plant. Other explanations for the high ash content could be the use of sulfur in distillation of second-generation bioethanol, in the case of residual biomass, and chemical contamination by pesticides or fertilizers during cultivation of sugarcane, in the case of sugarcane bagasse.

Globally, residual biomass performed superiorly to sugarcane bagasse in regards to its composition. This ingredient technically was comparable to other residues regular in the fabrication of wood and non-wood pellets for heating and power (Appendix A). This finding is timely and of great relevance to magnify and diversity the list of cost-effective additives for making high-performance fuel grade biosolids while not processing woody material. This may drive the global market of pellets towards the independence on residues of softwood and hardwood, coal and derivatives, like lignite, which is very pollutant [7,13,14].

#### 3.1.2. Distribution of Particles’ Size

The residual biomass relatively was higher in mass of particles in the optimal range for pelleting of 0.25–0.45 mm. The introduction of it as an additive into the co-pelletization with sugarcane bagasse should therefore enable grind particles to go smoothly through the machine to shape themselves into high-performance pellets. The finer and more homogeneous the material, the evener and more effective the feeding on the channel-forming die, as the feedstock gains in grindability, flowability, compressibility, and compactability [15]. These strengths are likely to optimize the pelleting throughout. Any improvement in the process would enhance pellet’s homogeneity, density, durability, and fuel power.

### 3.2. Quality and Potential Applications for the Hybrid Non-Wood Pellets

Pellets with the supportive material at the highest mass ratio ended up much more geometrically compact, due to great compressibility and compactability of the supplement. Shorter products often are suitable for easy handling, transportation, storage, and procedure. The ISO 17225-2, ISO 17225-6, and EM 14641-1 set the length to be in the range of 3.15–40 mm for both the residential and commercial non-wood pellets, while the USA PFI sets the length to be no great than 42 mm (Table 4). Therefore, both the composite and pure pellets would be suitable for both the residential and commercial applications, as their length fulfilled the requirements of these strict international standards for this variable. However, composite pellets almost entirely consisting of residual biomass should be very attractive over any other product to potential applications requiring high-technical quality solid biofuels to go smoothly. These hybrid materials were the most homogeneous and most compact. These features are of great relevance to reduce blockage of transport pipe in pneumatic systems [16].

Like length, density, durability, and HHV for the pellets with the supportive material at the highest mass ratio were also in agreement with the guiding values from the literature. Substantial enhancements in physicomechanical stability of these products were the proofs of this agro-industrial residue capable of successfully working like an organic binding agent in biomass. The addition enabled the particles to more efficiently diffuse into the voids, then properly fill them, thus reducing availability of structural pores. Practically, layers of finer particles in blends should move, push, and pull up to each other more consistently, thereby, forming themselves into denser products upon passing through the machine [13,17]. Any upgrade in the degree of compactness of composite pellets is likely to save space for the transportation and storage, make their logistic easier and cheaper, and optimize their combustion in heat-and-power units requiring dense products to perform finely.

Expressive gains in both the resistance to abrasion and durability should provide composite pellets mechanical strength enough to resist unpredictable conditions of transportation and storage, such as sudden changes in temperature and relative humidity of the air in containers and facilities. The great durability of these products may assist with solving very serious problems of environmental hygiene due to generation of dust and fines, off-gassing of toxic elements and self-firing. These phenomena can lead to lethal accidents, poisoning, and suffocation in workers and consumers by occupational exposure to deadly atmosphere during offloading transatlantic cargo and upon entering the setting, whether industrial and household [18]. The explanation for the significant durability of pellets almost entirely consisting of supportive material is the powerful bridging force and fine granulometry of the supplement. These characteristics enabled the pellets to more firmly support deformation by multiple falls, collisions, and shocks, and reduced the loss of dust and fines to the environment. The international standards do not set guiding value for the resistance to abrasion. However, this variable is of importance to predict mechanical strength and tensile resistance of the pellet. Resistance to abrasion varies proportionally with durability.

Improvements in both the gross heat of combustion and energy density in blends were due to substantial increments by the supportive material in contents of fixed carbon, lignin, C, and H. These properties make up the flammable and combustible portion of the biomass. High atomic ratios of C:O and H:O often yield larger amount of thermal energy, as the C–C and C–H bonds contain higher energy than C–O and H–O bonds, which are more hygroscopic and cause loss of energy by promoting vaporization on water [19]. Besides the excellent binding power, the supportive material also could perform like an energy-supplying additive in non-wood pellets due to its high gross heat of combustion and atomic ratios of C:O and H:O. This finding is timely and may be of great relevance to densification of carbon and concentration of energy in fuel grade biosolids from processing low-technical-quality raw material, such as the sugarcane bagasse.

Mechanical stability, fuel power, and oxidative potential of the pellet all depend on the proximal properties of the material [20]. The ISO 17225-2, EM 14641-1 and USA PFI set the water content to be no greater than 10% for residential non-wood pellets, while the ISO 17225-6 sets the water content to be no great than 12% for commercial products. Therefore, both the composite and pure pellets technically would fall in the strictest classes, as their water contents were in compliance with the requirements of international standards for this property. Almeida et al. [21] documented 77.25% volatile matter, 14.05% fixed carbon, and 8.70% for pellets from sugarcane bagasse, consistent with the proximal composition of pellets in this study, regardless of the type, whether composite and pure. The ISO 17225-2 and 17225-6 set the ash content to be no greater than 0.7 and 6% for the residential and commercial classes, respectively. Despite the fact that they fulfilled the requirements for the length and diameter, apparent density, HHV, and durability, pellets with the supportive material at the largest size of blending technically would not fall in the highest classes, as their ash content did not satisfy the requirements of international standards. Additionally, their sulfur content was beyond critical limit for this pollutant. Fuel power usually varies negatively with high contents of S and N. These elements are potential causes of corrosion of carbon-steel sheets in boilers and furnaces, and emission of SO_x_ and NO_x_ [22]. Further improvements in the preparation of blends to balance composite pellets to their contents of ash and S to ensure they are acceptable at the strictest grids of heat-and-power applications would therefore be mandatory.

Globally, supportive material of residual biomass from distillation of cellulosic ethanol enabled the hybrid non-wood pellets to be technically comparable and even superior to those from processing sugarcane bagasse [21], wheat straw [23], Moso bamboo [24], sewage sludge plus Chinese fir, Camphor and rice straw [6], residues of olive tree [25] and corncob [26], and other woody and non-woody wastes regular in the production of pellets on an industrial scale (Table 5).

## 4. Materials and Methods

### 4.1. Pre-Production

#### 4.1.1. Infrastructure

Laboratory of Machinery and Mechanics of the College of Agricultural and Technological Sciences, São Paulo State University (Unesp), Campus of Dracena, São Paulo, Brazil. The integrated set for pelleting comprised feeder silo, automatic pelletizer machine, with flat die, and vibrating screener with cooler (Figure 6), all supplied from Lippel^TM^. Specifications of the set are in Appendix A.

#### 4.1.2. Origination of the Starting Materials

The sugarcane bagasse, the base material, and residual biomass, the organic supportive material, both were from the same sugar-energy plant in Northeast Brazil. The supportive material referred to the residue from the stage of separation/distillation of cellulosic ethanol from processing sugarcane bagasse. The base material was from the procedure of extraction of stem-juice for production of sugar and first-generation bioethanol and cogeneration of heat, steam, and bioelectricity as well. The temperature and relative humidity of the air during initial storage of the materials in 10 kg polyethylene bags in laboratory to prevent them from experiencing adversities of surrounding weather agents which might influence their integrity were 25 ± 2.5 °C and 60 ± 5%, respectively.

##### Characterization

The apparent density documented the ratio of mass and volume of 0.05 kg sample in 0.25 L water in 1 L glass test tube, according to standard EN ISO 18847. The physicochemical characterization, in triplicate to ensure reproducibility, consisted of measuring/determining the contents of water, volatile matter, fixed carbon, and ash, through the proximate analysis; cellulose, hemicellulose, lignin, and extractives, through the structural analysis; C, H, O, N, and S, through the ultimate analysis; and higher heating value through the analysis of calorimetry, as the international norms summarized in Appendix A. The determination of distribution of particles’ size consisted of sieving 0.1 kg sample in set of stainless steel wire cloths with openings in the range of 0.15–0.85 mm, according to standard CEN/TS 15150.

### 4.2. Production

#### 4.2.1. Experiment

##### Planning

The experiment to test technical viability of blending sugarcane bagasse with residual biomass to develop high-performance hybrid non-wood pellets for heating and power used a completely randomized design. The mass ratios of sugarcane bagasse and the reinforcement were 3:1, 1:1, and 1:3 m m^−1^. Each test comprised ten replicates of 1 kg. Pellets purely consisting of sugarcane bagasse were used as the reference for eventual contrasting of performance.

##### Setup

This step started with conditioning the starting materials by drying in a horizontal airflow drying oven at 65 °C until 10–15% water, milling in Wiley knife mill, and then sieving in stainless steel wire cloths with openings of 0.25–0.45 mm [28]. The process of pelleting started with filling the hopper of the feeder silo with 10 kg material for each test. The feeder, working at 75 kg h^−1^, fed the pelletizer intermittently with the load, which slipped then through the channel-forming die underneath pressing rollers of the machine at 200 MPa and 125 °C to break itself into single pellets. The thresholds for the pressure and temperature strictly obeyed the ranges of 115–300 MPa [29] and 75–150 °C [28] for making pellets, whether woody and non-woody, optimally. The machine automatically ejected off the pellets from the flat die onto the vibrating screener at 1 g to prevent them from severe fractures by excessive vibration. The cooler made the heat to flash off from the mass of pellets, which were then stored in 1 kg polyethylene bags until further analytical procedures.

### 4.3. Post-Production

#### 4.3.1. Quantitative Analysis

The technical assessment of the batches of pellets consisted of measuring/determining mechanical, physical, and chemical properties. The international standards for drafting inference about their class and potential application were ISO 17225–2, ISO 17225–6, EN 1461–1, and USA PFI. These strictest normative set guiding values for the quality of non-wood solid biofuels, whether residential or commercial.

##### Physicomechanical

Length and diameter: the assessment of ***L*** and ***Ø***, both expressed in millimeters, consisted of measuring twenty-five randomly selected pellets, longitudinally and transversally, respectively, with a hardened stainless steel digital caliper (MrToolz, model FBA-ip54, 0.01-mm resolution).

Durability: the determination of ***δ***, consisted of dropping out the 0.1 kg sample from the tabletop at 1.85 mm high onto the floor for ten times, sieving the material in stainless steel wire cloth with openings of 0.25 mm, and then weighing it on an analytical digital scale of 0.0001 g resolution (Shimazu, model ATX–220, Barueri, SP, BR), by adapting the method of Gil et al. [14]; the calculation of this variable was through the ratio of final and initial mass of the sample, through Equation (1):(1)$$\delta \left(\%\right)=\left(\frac{m_{f}}{m_{i}}\right)100$$
where ***δ*** is the durability, ***m_i_*** is the initial mass in grams, and ***m_f_*** is the final mass in gram.

Resistance to abrasion: The determination of ***η***, expressed as percentage, consisted of shocking the 0.05 kg sample against itself and against the walls of the rotating chamber (0.2 m length by 0.2 m width by 0.1 depth), with 0.01 m high baffle across the sidewall, working at 50 rpm, clockwise, for 10 min, sieving the material in stainless steel wire cloth with holes of 0.85 mm, and then weighing it on an analytical digital scale, adapting the method of Hosseinizand et al. [27]; the calculation of this variable was through the ratio of final and initial mass of the sample through Equation (2).
(2)$$\eta \left(\%\right)=\left(\frac{m_{f}}{m_{i}}\right)100$$
where ***η*** is the resistance to abrasion, ***m_i_*** is the initial mass in gram, and ***m_f_*** is the final mass in gram.

##### Physicothermal

Apparent density: the ***ρ***, expressed as kilogram per cubic meter, was the mass to volume ratio of 0.1 kg sample in 0.5 L water in 1 L glass test tube, adapting the method of Liu et al. [24].

Higher heating value: the HHV, expressed as mega Joule per kilogram, was the amount of energy releasing from burning down 0.1 kg sample on an isothermal calorimeter (IKA, model C200, Campinas, SP, Brazil) containing oxygen and helium as tracer and carrier gases.

Energy density: the ***ε***, expressed and giga Joule per cubic meter, was ***ρ*** vs. HHV.

##### Physicochemical

The assessment of chemical quality consisted of processing 0.5 kg sample, in triplicate, through the same analytical procedures for the characterization of the starting materials, except structural analysis. The chemical analysis will generate useful graphs for the effect of the supportive material on the flammable and combustible properties of hybrid non-wood pellets at the molecular level, mainly with respect to the contents of C, H, and O, and their atomic ratios.

### 4.4. Data Analysis

For the formal analysis of the data, Shapiro–Wilk and Bartlett procedures were performed to check normalcy and homoscedasticity, respectively. The one-way analysis of variance to test the significance of effect of share of supportive material in the blend with sugarcane bagasse on the technical performance of composite pellets was carried out. The regression analysis to fit the data was carried out. The metric for the analysis of accuracy of first-power regression models was coefficient of determination (*r*^2^). Another method of applying non-traditional mathematics to assist tracking linear relationships between characteristics of the ingredients and quality of products encompassed the canonical correlation analysis. The software was R-project [30], which complies and runs on several platforms. This multi-paradigm programming language provides user-friendly environment, packs, and libraries for statistical computing and graphics.

## 5. Conclusions

Preliminary evidence of lignocellulosic supportive material at 75% total mass of the blend capable of highly improving flammable/combustible properties at the molecular scale, physicomechanical stability, and fuel power of hybrid pellets exist. Pellets consisting almost entirely of residual biomass can yield 53.50% carbon, 7.80% hydrogen, 99.55% durability, and 31.20 MJ kg^−1^ energy density. This significant durability may assist solving very serious problems of environmental hygiene in workers and consumers by occupational exposure to generation of dust and fines, off-gassing and self-firing in ships and trucks’ containers and indoor facilities with no systematic heating-up or ventilation. Practically, hybrid non-wood pellets should be of relevance for industrial applications, such as coal-firing power stations. Further improvements in the contents of ash and sulfur to ensure composite non-wood pellets technically viable to power the strictest residential and commercial heat-and-power units would be, therefore, necessary. To optimize co-pelletization, future researches will combine thermochemical pretreatment, supplementation, and operational conditions on the machine to fit the hybrids non-wood pellets to the guiding limits of ash and sulfur from the international standards, ISO 17225-2 and ISO 17225-6, EM 14641-1, USA PFI. The focus will be to provide them combination of efficiency and reliability for thermal conversion equipment and comfort for stakeholders, whether manufacturers, suppliers, and consumers.

## Figures and Tables

**Figure 1 molecules-25-04219-f001:**
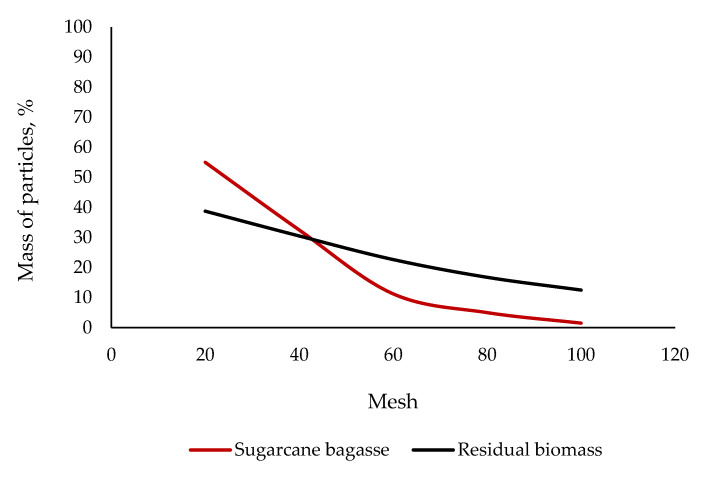
Distribution of particles’ size of the ingredients for making hybrid non-wood pellets. The flatter the curve is, the more homogeneous the feedstock is, as it gains in regularity of size and shape of particles. The steeper the curve is, the greater the probability is that the material will become heterogenous, as the range of size of particles, from fine to coarse, is likely to be significantly large.

**Figure 2 molecules-25-04219-f002:**
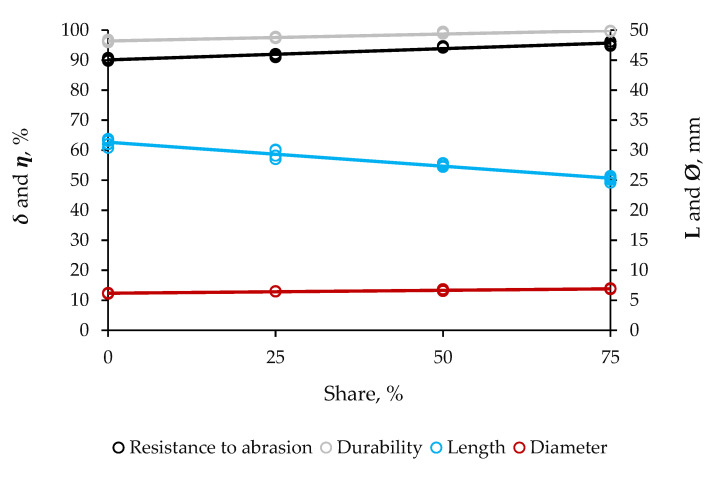
Effect of share of residual biomass from distillation of cellulosic bioethanol in the blend with sugarcane bagasse on the physicochemical quality of hybrid non-wood pellets. The steeper the line, the larger the probability of the factor to becoming the cause of significant variability, as it gains in linearity; length (***L***), diameter (***Ø***), durability (***δ***), and resistance to abrasion (***η***).

**Figure 3 molecules-25-04219-f003:**
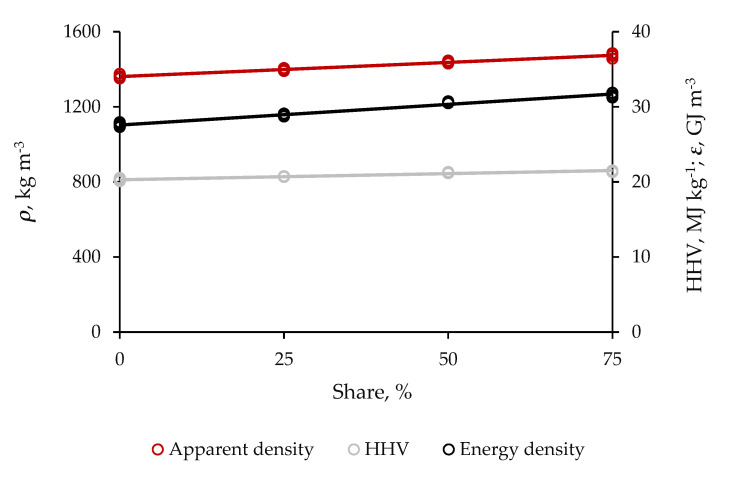
Effect of share of residual biomass from distillation of cellulosic bioethanol in the blend with sugarcane bagasse on the physicothermal quality of hybrid non-wood pellets. The steeper the line is, the larger the probability of the factor becoming the cause of significant variability is, as it gains in linearity; apparent density (***ρ***), higher heating value, (HHV) and energy density (***ε***).

**Figure 4 molecules-25-04219-f004:**
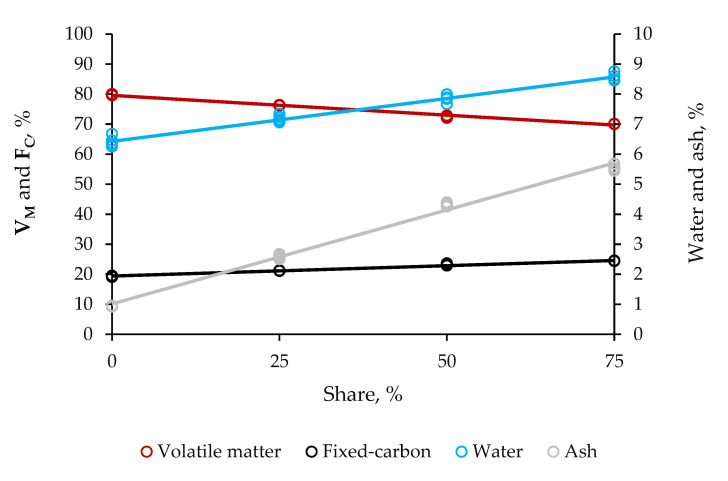
Effect of share of residual biomass from distillation of cellulosic bioethanol in the blend with sugarcane bagasse on the proximal properties of hybrid non-wood pellets. The steeper the line is, the larger the probability of the factor becoming the cause of significant variability is, as it gains in linearity; volatile matter (V_M_) and fixed carbon (F_C_).

**Figure 5 molecules-25-04219-f005:**
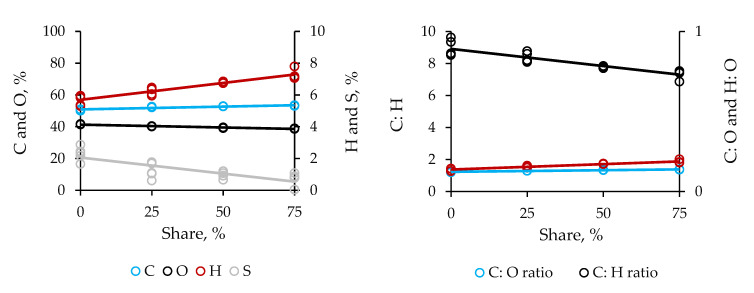
Effect of share of residual biomass from distillation of cellulosic bioethanol in the blend with sugarcane bagasse on the elemental properties of composite pellets. The steeper the line is, the larger the probability of the factor to becoming the cause of significant variability is, as it gains in linearity.

**Figure 6 molecules-25-04219-f006:**
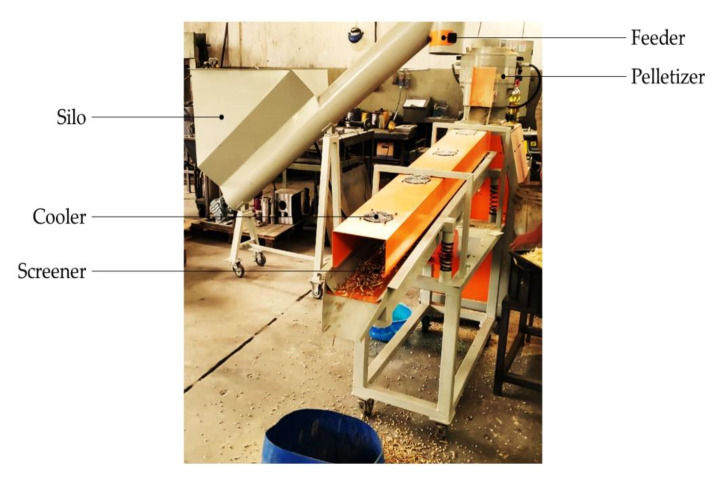
Pilot-scale integrated set for pelleting.

**Table 1 molecules-25-04219-t001:** Physicothermal properties of the ingredients for making hybrid non-wood pellets.

Property	Ingredient	
	Sugarcane Bagasse	Residual Biomass
	Proximal	
Water, %	13.80	12.20
Volatile matter, %	76.35	69.85
Fixed carbon, %	15.40	24.30
Ash, %	8.25	5.85
	Structural	
Cellulose, %	43.80	41.30
Hemicellulose, %	27.70	28.70
Lignin, %	20.95	22.70
Extractive, %	7.55	7.30
	Elemental	
C, %	44.95	53.05
H, %	5.95	6.05
O, %	47.85	39.75
N, %	-	-
S, %	1.20	1.30
Atomic C: O ratio	0.90	1.30
Atomic H: O ratio	0.10	0.15
Atomic C: H ratio	7.55	8.75
	Physicothermal	
Apparent density, kg m^−3^	560.05	590.15
Higher heating value, MJ kg^−1^	17.65	21.85
Energy density, GJ m^−3^	9.90	12.90

**Table 2 molecules-25-04219-t002:** Analysis of variance for the effect of blending of residual biomass from distillation of cellulosic bioethanol on the quality of hybrid non-wood pellets.

Property	Assumption		Source of Variation	
	Normalcy	Homogeneity	Blending	Coefficient of Variation, %
	*p*-Value	*p*-Value	*F*-Value	
Physicomechanical				
Length	0.35 *	0.50 *	18.15 *	1.05
Diameter	0.70 *	0.30 *	6.05 *	1.80
Resistance to abrasion	0.05 *	0.55 *	10.20 *	1.25
Durability	0.10 *	0.70 *	90.05 *	0.25
Physicothermal				
Apparent density	0.85 *	0.05 *	44.50 *	0.20
Higher heating value	0.65 *	0.65 *	15.05 *	1.95
Energy density	0.70 *	0.70 *	27.30 *	1.05
Physicochemical				
Water	0.25 *	0.85 *	105.15 **	0.70
Volatile matter	0.40 *	0.55 *	135.55 **	0.70
Fixed carbon	0.45 *	0.40 *	57.60 **	1.05
Ash	0.05 *	0.10 *	496.05 **	0.95
C	0.45 *	0.20 *	221.20 **	0.15
H	0.95 *	0.15 *	95.35 *	0.80
O	0.55 *	0.55 *	105.45 **	0.75
N	0.50 *	0.15 *	0.05	0.05
S	0.50 *	0.80 *	25.75 *	2.10
Atomic C:O ratio	0.70 *	0.40 *	230.95 **	0.80
Atomic H:O ratio	0.50 *	0.75 *	195.25 **	0.45
Atomic C:H ratio	0.90 *	0.25 *	10.50 *	0.70

Significant code: ** *p* < 0.01; ** p* < 0.05.

**Table 3 molecules-25-04219-t003:** Parameters and goodness-of-fit for the first-power models for the effect of blending of residual biomass on the quality of hybrid non-wood pellets.

Property	Parameter		*r* ^2^
	*β*_0_, Intercept	*β*_1_, Blending	
	Physicomechanical		
Length, mm	6.175 *	−0.010 *	0.915
Diameter, mm	31.330 *	0.080 *	0.955
Resistance to abrasion, %	96.390 *	0.045 *	0.950
Durability, %	90.095 *	0.075 *	0.930
	Physicothermal		
Apparent density, kg m^−3^	1361.100 **	1.510 **	0.970
Higher heating value, MJ kg^−1^	20.280 *	0.015 *	0.940
Energy density, GJ m^−3^	27.570 **	0.055 **	0.980
	Physicochemical		
Water, %	6.425 **	0.030 **	0.975
Volatile matter, %	79.560 **	−0.130 **	0.990
Fixed carbon, %	19.400 **	0.070 **	0.985
Ash, %	1.010 **	0.060 **	0.995
C, %	50.920 *	0.035 *	0.945
H, %	5.695 *	0.210 *	0.875
O, %	41.330 *	−0.035 *	0.945
S, %	2.060 *	−0.020 *	0.645
Atomic C:O ratio	1.230 *	0.005 *	0.925
Atomic H:O ratio	0.135 *	0.0005 *	0.910
Atomic C:H ratio	8.915 *	0.0215 *	0.805

Significant code: *** p* < 0.01; ** p* < 0.05.

**Table 4 molecules-25-04219-t004:** Potential applications for the hybrid non-wood pellets from co-pelletization of sugarcane bagasse with residual biomass from distillation of cellulosic ethanol.

Property	Pellet		Norm			
	Composite	Pure	ISO 17225-2	ISO 17225-6	EN 1461-1	USA PFI
Potential application	Industrial	Industrial	Residential	Industrial		
Apparent density, kg m^−3^	1405.25–1485.10	1375.15	≥600	≥600	≥600	608.7–746.9
Heating value, MJ kg^−1^	20.70–21.45	20.50	≥16.5	≥14.5	≥16.5	-
Length, mm	25.70–30.10	31.90	3.15–40	3.15–40	3.15–40	≤42
Diameter, mm	6.50–6.85	6.05	6, 8, 12 ± 1	6–10	6, 8, 10 ± 1	5.84–7.25
Durability, %	97.10–99.95	95.90	≥97.5	≥97.5	≥96.5	≥95
Water, %	7.05–8.75	6.70	≤10	≤12	≤10	≤10
Ash, %	2.65–5.45 ^†^	0.90 ^†^	≤0.7	≤6	≤1	≤2
N, %	0.00	0.00	≤0.3	≤1.5	≤0.5	-
S, %	0.90–1.05 ^†^	2.45 ^†^	≤0.04	≤0.2	≤0.05	-

^†^ Limiting for the potential application of pellets in the strictest grids of residential or commercial heat-and-power units.

**Table 5 molecules-25-04219-t005:** Relative quality of hybrid non-wood pellets from pressing sugarcane bagasse with residual biomass from distillation of cellulosic bioethanol.

Feedstock	Quality										Reference
	Physicothermal			Physicomechanical			Chemical				
	*ρ*, kg m^−3^	HHV, MJ kg^−1^	*ε*, GJ m^−3^	*L*, mm	*Ø*, mm	*δ*, %	Water, %	Ash, %	N, %	S, %	
Sugarcane bagasse plus residual biomass	1485.10	21.45	31.85	30.10	6.85	99.95	8.75	5.45	0.00	1.05	
Sugarcane bagasse	1375.15	20.50	28.20	31.90	6.05	95.90	6.70	0.90	0.00	2.45	
Sugarcane bagasse	726.3	16	-	22.7	9.7	98.2	-	8.7	0.3	0.02	Almeida et al. [21]
Olive leaves	<1000	19.65	<19.65	12.3	6	88.6	-	-	-	-	Garcia-Maraver et al. [25]
Olive prunings	>1000	-	-	24	6	-	-	-	-	-	
Olive wood	>1000	17.5	>17.5	28.7	6	91.7	-	-	-	-	
Bamboo plus rice straw	900–1350	15.4–18.25	13.85–24.6	13.6	6	94.1–99	-	2–16	-	-	Liu et al. [24]
Chinese fir	1110	-	-	-	7	-	-	-	-	-	Jiang et al. [6]
Camphor	1105	-	-	-	7	-	-	-	-	-	
Rice straw	1180	-	-	-	7	-	-	-	-	-	
Treated wheat straw	969–1035	-	-	-	6	-	-	-	-	-	Gao et al. [23]
Corn stover with starch	-	-	-	-	6	88–98.8	11–16	-	-	-	Djatkov et al. [26]
Corn cob with starch	-	-	-	-	6	77.9–99.2	8–15	-	-	-	
Microalgae	1192–1229	27.8	-	-	6	82–96.5	-	2.5	-	-	Hosseinizand et al. [27]
Sawdust	817–1038	19.4	-	-	6	29–85.8	-	0.1	-	-	
Microalgae plus sawdust	1155–1207	25.3	-	-	6	72.4–97.7	-	1.7	-	-	

Length (***L***), diameter (***Ø***), durability (***δ***), apparent density (***ρ***), higher heating value (HHV), energy density (***ε***).

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
