# Peer review of "Second-Generation Lignocellulosic Supportive Material Improves Atomic Ratios of C:O and H:O and Thermomechanical Behavior of Hybrid Non-Woody Pellets"

_molecules, 2020, doi:10.3390/molecules25184219_

Round 1

Reviewer 1 Report

Dear Authors,

I have reviewed the paper "Second-Generation Lignocellulosic Supportive Material Can Spectacularly Improve Atomic Ratios of C:O and H:O, and Thermomechanical Behavior of Hybrid Non-Woody Pellets". The aims of the paper are germane with Molecules journal, in this form of article fits with the international scientific standards, although some flaws are present, for these reasons changes are needed. The paper is written with an appropriate technical English level. The contribution of this paper to the scientific knowledge in the present form could be considered good but some corrections also in formatting are necessary. I suggest the corrections in the comments and also in the files attached.

I suggest the corrections in the file attached and, in the comments, below:

  • The paper provides an interesting study on blended pellet production. However, there are several flaws. First the tone in which it is written is not scientific but actually too much autocelebrative, terms as "spectacularly" or "impressively" are not suitable to a scientific context.
  • Materials and Methods are put almost at the end of the paper while they must be moved after Introduction.
  • There is the strong need to resume the main results and findings with table put in the main text.
  • Introduction: In my opinion it is necessary at the end of this chapter to add a period with the main focuses/aims of this paper, also in order to have a better linkage with the conclusions.
  • References: please attention in formatting the papers correctly, mainly referred to journal abbreviation.

Reviewer 2 Report

An article is too long, it contains a lot of general information, distant from the subject of the research presented. The article should be considerably shortened. The information provided must focus solely on the purpose of the research, the results achieved, the research materials and methods relating solely to the research conducted.

The purpose of the Introduction chapter is to clearly convey the purpose of the research and to describe the current state of knowledge on the issue under study. After reading the Introduction in this article, we don't know what the authors mean, what exactly they intend to research and for what purpose. The knowledge provided in this chapter is too general, the aim of the research is not presented.

The Results section should be focused only on the presentation of tables with data from the study of materials, including data included in supplementary materials, and on graphs of the results. There is too much descriptive information.

The discussion chapter also contains too much background information. The content should briefly evaluate the results obtained and compare them with the results of similar studies by other scientists.

The authors, shortening the article and specifying the information contained therein only to the phenomena studied, should consider the legitimacy of such a large amount of the cited items of literature - 129.

Round 2

Reviewer 1 Report

Dear Authors,

I have reviewed the paper "Second-Generation Lignocellulosic Supportive Material Can Spectacularly Improve Atomic Ratios of C:O and H:O, and Thermomechanical Behavior of Hybrid Non-Woody Pellets". The aims of the paper are germane with Molecules journal, in this form of article fits with the international scientific standards, in the present form the previous flaws present have been corrected. The paper is written with an appropriate technical English level. The contribution of this paper to the scientific knowledge in the present form could be considered good. Only one minor concern, I'm sorry you didn't try to slightly implement some introductory secondary aspects by considering new papers, it was not a fundamental thing but it would certainly have made this interesting article more multidisciplinary and of wider interest. I congratulate you on what has been done.

Reviewer 2 Report

This article in its current form, after the changes made, has become more reader-friendly.

The research results included in the manuscript text provide a lot of interesting data. Their interpretation became more understandable than in the previous version of the manuscript.

Despite the fact that in this version of the article the same results as in the previous one were presented, the rewording of the article gave it more scientific power.